# Impacts of Fracture Roughness and Near-Wellbore Tortuosity on Proppant Transport within Hydraulic Fractures

**Di Wang [1,2], Bingyang Bai [3], Bin Wang [3], Dongya Wei [3] and Tianbo Liang [3,*]**

1 State Key Laboratory of Shale Oil and Gas Enrichment Mechanisms and Effective Development, Beijing 100083, China; wangdi2018.syky@sinopec.com
2 National Energy Shale Oil Research and Development Center, Beijing 100083, China
3 Unconventional Petroleum Research Institute, China University of Petroleum at Beijing, Beijing 102249, China; 13717591719@163.com (B.B.); 2020211834@student.cup.edu.cn (B.W.); weidydr@cnpc.com.cn (D.W.)
* Correspondence: btliang@cup.edu.cn

**Abstract:** For unconventional reservoir hydraulic fracturing design, a greater fracture length is a prime factor to optimize. However, the core observation results from the Hydraulic Fracturing Test Site (HFTS) show that the propped fractures are far less or shorter than expected, which suggests that the roughness and tortuosity of hydraulic fractures are crucial to sand transport. In this study, a transport model of sands is first built based on experimental measurements on the height and transport velocity of the sand bank in fractures with predetermined width and roughness. The fracture roughness is quantified by using the surface height integral. Then, three-dimensional simulations are conducted with this modified model to further investigate the impact of tortuous fractures on sand transport, from which a regression model is established to estimate the propped length of hydraulic fractures at a certain pumping condition. The experiment results show that the height of the sand bank in rough fractures is 20–50% higher than that in smooth fractures. The height of the sand bank decreases with the reduction in slurry velocity and increases with the increase in sand diameter. Sand sizes do little effect on the transport velocity of the sand bank, but the increase in slurry velocity and sand volume fraction can dramatically enhance the migration velocity of the sand bank. The appearance of tortuous fractures decreases the horizontal velocity of suspended particles and results in a higher sand bank compared with that in straight fractures. When the sand bank reaches equilibrium at the tortuous position, it is easy to produce vortices. So, there is a significant height of sand bank change at the tortuous position. Moreover, sand plugging can occur at the entrance of the fractures, making it difficult for the sand to transport deep into fractures. This study explains why the propped length of fractures in HFTS is short and provides a regression model that can be easily embedded in the fracturing simulation to quickly calculate dimensions of the propped fractures network to predict the length and height of propped fractures during fracturing.

**Keywords:** proppant transport; fractures roughness; tortuous fractures

## 1. Introduction

With the decrease in conventional oil and gas resources, people turn their attention to the exploitation of unconventional resources. Presently, the exploitation of most unconventional resources benefited from the development of horizontal wells and multi-stage fracturing technology. In tight oil and gas reservoirs, the propped fracture is of great importance to provide flow channels for oil and gas production. In addition, the distribution of proppant significantly influences the conductivity of fractures. Thus, it is important to investigate the sand bank dune form under different construction conditions. Two main factors affect the transportation and distribution of proppant in fractures. One is the properties of the fractures including their width, height, roughness, and tortuosity, and the other is the pumping schedule including the proppant density and diameter, the rheology of fracturing fluid, the slurry velocity, and the volume fraction of proppants.

Numerous experimental and simulation studies have been conducted to investigate the transport and distribution of proppant in fractures. The motion of proppant in the fractures can be divided into the following two aspects: the settling of proppant in the vertical direction and transport in the horizontal direction [1]. It can be inferred that investigating the law of proppant settling is of great significance [2]. Kern et al. [3] found that the proppant settling velocity decreases with the particle diameter closing to the fracture width. Shrivastava and Sharma [4] found that the smaller particles have good suspension and can transport deeper into the natural fractures. Harrington et al. [5] put forward a modified stokes formula to describe proppant settling in non-Newtonian fluid. The proppant settling velocity will decrease when the concentration of proppant increases when the particles are not clustered [6]. Gadde et al. [7] investigated proppant settling in rough fractures and found that the proppant settling velocity decreased when compared to the smooth fracture. There is a strong positive correlation between proppant placement concentration and local fracture roughness [8]. Debotyam et al. [9] found that during the fracture propagation period, the stress at the boundary and the lithology change have a great influence on the proppant distribution.

As proppants flow into hydraulic fractures, they continuously precipitate from the slurry and form a sand bank, whose equilibrium height is determined by Kern et al. [3] and Patankar et al. [10]. Wang et al. [11] used the experimental data of STIM-LAB to establish bi-power law correlations for the bed load transport of slurries. Gadde et al. [7] put forward a modified correlation of stokes considering fluid viscosity and fracture width. They found that the propped length of fractures increases when incorporated into UTFRAC-3D to predict the proppant distribution in fractures. Dontsov [12] studied the gravity-dominated proppant settlement in hydraulic fractures. The proppant distribution is uneven within the tortuous fractures in shale reservoirs [13]. The proppant will occupy a greater coverage area when transported in inclined fractures [14]. After that, Chun et al. [15] found that a small percentage of proppant can travel into the bedding plane for T-shape fractures. Fernández et al. [16] studied the velocity field of proppant transport in a large laboratory flat vertical fractures by using Particle Image Velocimetry. The roughness of the face of fractures can increase the interaction between particles and rock, which increases the uncertainty of flow [17]. Huang et al. [18] also investigated the proppant distribution in rough fractures and found that the form of sand bank is not always the same as that in smooth fractures. Compared with smooth fractures, there will be more proppant deposited in rough fractures at a high pumping rate. [18,19]. However, the roughness of fracture surfaces is not quantitatively used in these studies. Presently, the quantification of fracture roughness is mainly as follows. Authors [20] proposed the joint roughness coefficient (JRC) to characterize the roughness of different types of rock surface. The fractal dimension is another method to quantify the roughness of fractures, which can be obtained by the variogram analysis, power spectral density, and triangular prism. In addition, a relatively simple method to quantify the fracture roughness is the surface integral ratio, which is the ratio between the total fracture surface area and the projection area of rough surface [21].

The simulation of proppant transport can be mainly divided into two categories. One is the Eulerian–Eulerian method and the other is the Eulerian–Lagrangian method. The most common method used to simulate the proppant transportation in filed scale is the Eulerian–Eulerian method. The Eulerian–Eulerian method refers to the use of the mass conservation equation to calculate fluid and proppant in the Eulerian grid. The transport of proppant is assumed as the change of proppant volumetric concentration. It may be difficult to capture every particle movement, but it saves a lot of calculation time. The Eulerian–Lagrangian method is another method to simulate proppant transport and distribution. The Discrete Particle Method (DPM) and Computational Fluid Dynamics-Discrete Element Method (CFD-DEM) are the two mainly used Eulerian–Lagrangian methods. The DPM model is suitable to solve proppant transport with low concentration. CFD-DEM can model every particle movement and high concentrations of proppant transport. Due to tracking the movement of each particle, CFD-DEM is more time-costing and has a large amount of

calculation. This method can describe the particle transport accurately but it is difficult for field scale simulation. A dimension reduction strategy coupled with the Eulerian–Eulerian method is used to investigated the proppant transportation in filed scale by Hu et al. [1]. Suri et al. [22] investigated the proppant transport in fracture roughness and found the roughness of fractures can provide an additional force for proppant transport in fractures for a long distance. Baldini et al. used the CFD-DEM method to study the proppant transport and sand bank form in a planar vertical fracture [23]. Han et al. used LBM-DEM to model the granular flow in porous media [24].

From the previous experiments and simulation results, it is known that the injected proppant can form a regular sand dune in the fractures and transport deep into the fractures as continuously injected slurry. The distribution of proppant in real fractures remains unknown.

The Hydraulic Fracture Test Site project aimed to recover cores from a stimulated volume, making a further observation of the proppant distribution in the real fractures. However, the core results show an unexpected distribution of proppant. Many of the cores collected held little sand [25]. Moreover, the distribution of proppant in the fractures was nonuniform. Many of the proppants were found in the fractures near wellbore and little proppant was discovered in the secondary fractures [9].

Although a large amount of numerical and experimental work is conducted to investigate the proppant transportation and distribution in fractures, including the proppant-bed equilibrium height, the proppant-bed length is seldom studied. As for the proppant distribution in fractures, the sand bank height is a significant parameter to depict the vertical sand coverage. However, the propped length of fractures in roughness field scale fractures is unclear. In this study, a regression model is proposed to investigate the proppant transport and distribution in rough fractures, which gives a prediction in field scale. In addition, a tortuous fracture model is applied to explain the unevenness of proppant distribution.

## 2. Model Buildup

In this study, the principal purpose is to explain the transportation and distribution of proppant in rough and tortuous fractures. The Eulerian–Lagrangian method coupled with the Dense Discrete Particle Model (DDPM) is applied to solve this problem of granular flow with a high volume fraction of particles.

### 2.1. Experiment Setup

Firstly, a large transparent fracture-flow model with smooth fracture surfaces is used to observe the formation and transport of a sand bank and compared with ones in published studies. This model is composed of two parallel Plexiglas plates, a centrifugal pump (with the maximum flow rate of 8 m$^3$/h), an agitator tank (with a maximum volume of 200 L), and a flowmeter (with a maximum flow rate of 8 m$^3$/h), as shown in Figure 1. The transparent fracture-flow model is used to observe the sand transport in a narrow and flat fracture, whose results can be used to validate the numerical model in this study. In the transparent fracture-flow experiments, silica sand and tap water with a certain sand volume fraction are continuously injected into a fracture with dimensions of 1.5 m (length) × 0.3 m (height) × 0.004 m (width).

The dimensionless number, Reynolds number, shown in Equation (1) of the particle is used to describe the granular flow, since the Reynolds number of particle flow in the fracture is proportional to the velocity of particles. There are generally 20 to 30 stages in the field fracturing construction. Each stage has about 3 to 6 perforation clusters and has about 5–10 m$^3$/min liquid inflows. In order to ensure the maximum similarity of the flow, the velocity used is shown in Table 1. Other parameters can be referred in Table 2, shown below.

$$\mathrm{R}_e = \frac{\rho v d}{\mu} \tag{1}$$

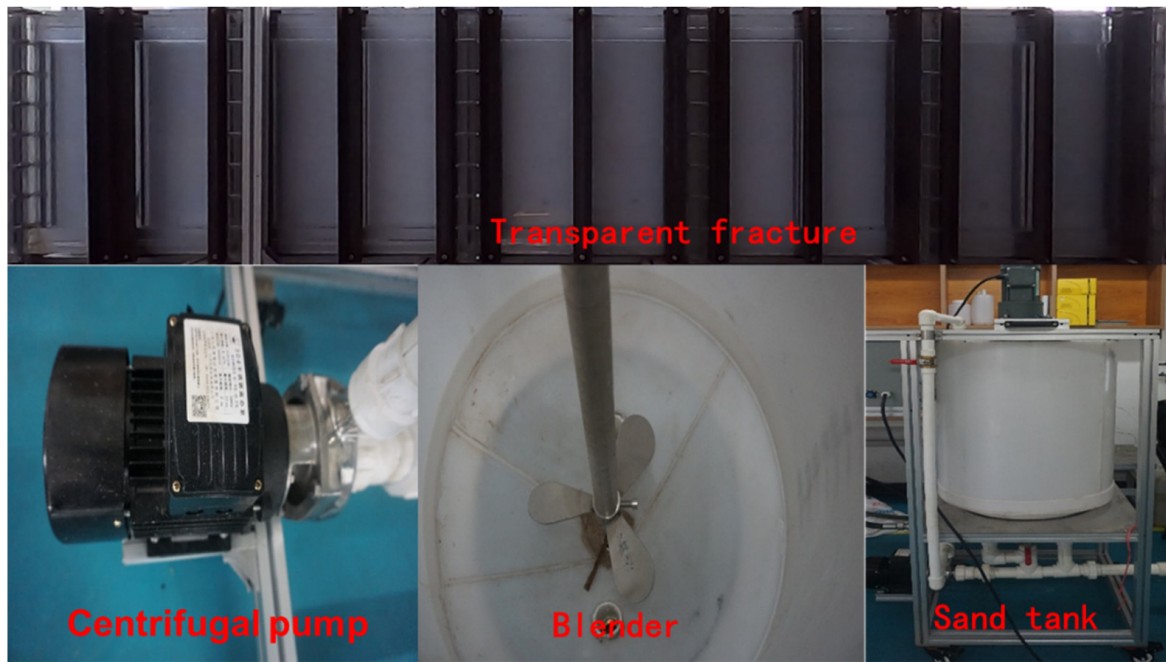

**Figure 1.** The transparent fracture-flow model.

**Table 1.** Comparison of flow rates between field and lab.

| | Volumetric Flow Rate (m³/min) | Fracture Height (m) | Fracture Width (mm) | Flow Rate (m/min) | Particle Reynolds Number | Fluid Reynolds Number |
|---|---|---|---|---|---|---|
| Field Scale | 5–10 | 10–60 | 2–4 | 3.5–167 | 30–5337 | 457–21,929 |
| Lab Scale | 0.01–0.25 | 0.3 | 4 | 8.3–208 | 75–6590 | 1096–27,412 |

**Table 2.** Parameters of experiments and simulations.

| Parameters | Values |
|---|---|
| Slurry velocity, m/min | 12, 24, 36 |
| Sand diameter, mm | 0.208, 0.325, 0.425, 0.739 |
| Fracture width, mm | 4 |
| Fluid viscosity, Pa·s | 0.001 |
| Real sand density, kg/m³ | 2600 |
| Sand bulk density, kg/m³ | 1500 |
| Volume fraction of sand | 7.3% |

The $R_e$ is Reynolds number. The $\rho$ and $v$ are the density and velocity of fluid. The $d$ and $\mu$ are the hydraulic diameter of the fracture and the viscosity.

The experiments are conducted using 20/40 mesh sand, with 7.3% sand and a slurry velocity of 12 m/min, 24 m/min, and 36 m/min. As shown in Figure 2, sands precipitate from the slurry and gradually form a sand bank; the height of the sand bank within the fractures increases with time until the equilibrium height is reached, which is similar to those reported in previous studies [11,26].

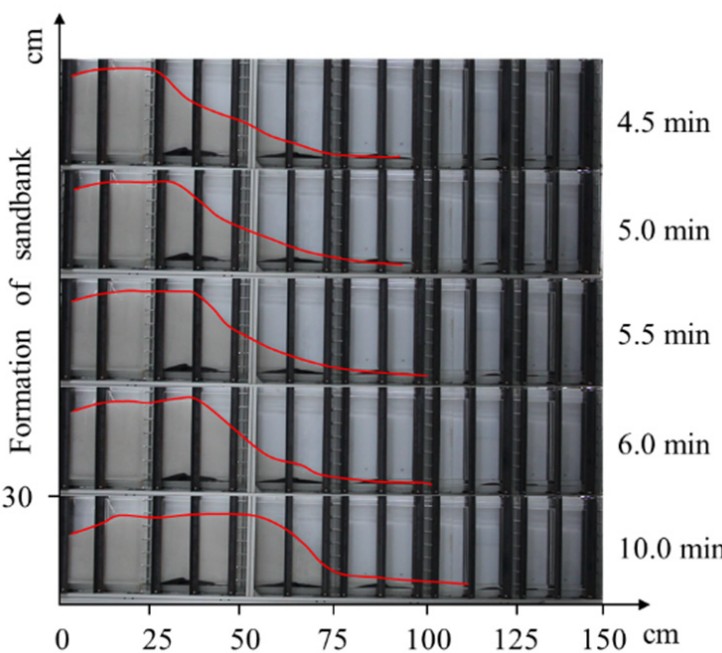

**Figure 2.** Comparison of sand bank, slurry velocity 0.12–12 m/min.

### 2.2. Simulation Setup

The Eulerian–Eulerian method and Eulerian–Lagrangian method are the two main methods applied to investigate sand transport and distribution in fractures. Sand and fluid are considered as two phases, respectively, in the Eulerian–Eulerian method, and both the fluid and sand are solved in Eulerian grids. The Eulerian–Lagrangian method can be roughly divided into two methods including the Discrete Particle method (DPM) and Computational Fluid Dynamics-Discrete Element Method (CFD-DEM). In the DPM method, the sands with similar properties are packed into parcels, which can dramatically decrease the calculational cost. In the CFD-DEM method, the movement of each individual sand is calculated in the Lagrangian method, which significantly increases the computational cost. In this study, the Eulerian–Lagrangian method with the Dense Discrete Phase Model (DDPM) model is chosen to study the sand transport and sand bank within a fracture. Some of the assumptions in this model are as follows.

Fluid is assumed as a continuous phase and is solved in the Eulerian grid. Sand is assumed as a discrete phase and is tracked by a Lagrangian approach and mapped back to the Eulerian grid. The force between sands is solved by the kinetic theory of granular flow model. During the sand injection, fracture propagation is not considered.

In the simulation, the same parameters are used.

Then, the appropriate boundary conditions and calculation parameters need to be selected. The inlet boundary is set as the velocity inlet, in which the velocity is set, as shown in Table 2. The outlet boundary is set as the pressure outlet, which is equal to the atmospheric pressure. Four different mesh sizes are injected using the Rosin–Rammler particle distribution. The particle size in the following text represents the largest particle size in this distribution. The front, behind, top, and bottom walls are specified as no-slip stationary walls. As the slurry is incompressible, the pressure-based solver is selected. The transient state is used to understand the sand transport. The Shear Stress Transport (SST) k-ω turbulent model is used. The SST k-ω model is a two-equation eddy-viscosity model and combines the standard k-ω turbulent model in the boundary layer with the standard k-ε turbulent model. The advantages of using the SST k-ω model is that it also provides excellent results in adverse pressure gradients and separating flow [27]. The Gidaspow et al. [28] model is used to simulate the viscosity of the granular phase. The time step used in the simulation is 0.001 s. The phase-coupled SIMPLE algorithm is used

to solve the pressure-velocity coupling [29]. The second-order upwind scheme is used to solve the discretization of momentum, volume fraction, and turbulent kinetic energy.

The simulation model is conducted to understand the effect of fracture roughness on sand transport and distribution. The transport velocity and equilibrium height of the formed sand bank within the fractures are compared among cases with different slurry velocity, sand sizes, sand volume fraction, and fracture surface roughness, shown in Table 3.

**Table 3.** Parameters in simulation cases.

| Parameters | Values |
| --- | --- |
| slurry velocity, m/min | 12, 24, 36, 66, 96, 126 |
| sand diameter, mm | 0.208, 0.325, 0.425, 0.739 |
| fracture width, mm | 4 |
| sand volume fraction | 4.7%, 7.3%, 10% |
| Ra (fracture roughness) | 0.39, 0.59, 0.79 |

To solve the equations, the inlet boundary is chosen as the velocity inlet where the slurry is injected, and the outlet boundary is set as one atmospheric pressure. The roughness of fracture surfaces is controlled by adjusting Ks, which is embedded in the boundary condition. DDPM is a well-established model and a brief introduction is as follows.

The continuity equation can be expressed as (Gidaspow) [28]

$$\frac{\partial}{\partial t}(\alpha_f \rho_f) + \nabla(\alpha_f \rho_f \vec{v_f}) = 0 \tag{2}$$

$$\frac{\partial}{\partial t}(\alpha_s \rho_s) + \nabla(\alpha_s \rho_s \vec{v_s}) = 0 \tag{3}$$

The $\alpha_f$ and $\alpha_s$ are the volume fraction of fluid and solid, respectively. The $\rho_f$ and $\rho_s$ are the density of the fluid and solid, respectively. The $v_f$ and $v_s$ are the velocity of fluid and solid, respectively.

The motion equation can be expressed as (Gidaspow) [28]

$$\frac{\partial}{\partial t}(\alpha_f \rho_f \vec{v_f}) + \nabla(\alpha_f \rho_f \vec{v_f}\vec{v_f})$$
$$= -\alpha_f \nabla p + \nabla \cdot \overline{\overline{\tau_f}} + \alpha_f \rho_f \vec{g} + \beta(\vec{v_s} - \vec{v_f}) \tag{4}$$

$$\frac{\partial}{\partial t}(\alpha_s \rho_s \vec{v_s}) + \nabla(\alpha_s \rho_s \vec{v_s}\vec{v_s})$$
$$= -\alpha_s \nabla p + \nabla \cdot \overline{\overline{\tau_s}} + \alpha_s \rho_s \vec{g} + \beta(\vec{v_f} - \vec{v_s}) \tag{5}$$

$$\overline{\overline{\tau_f}} = \alpha_f \mu_f (\nabla \vec{v_f} - \nabla \vec{v_f}^T) - \frac{2}{3}\alpha_f \mu_f (\nabla \cdot \vec{v_f})I \tag{6}$$

$$\overline{\overline{\tau_s}} = \mu_s (\nabla \vec{v_s} - \nabla \vec{v_s}^T) + (\lambda_s - \frac{2}{3}\mu_s)(\nabla \cdot \vec{v_s})I \tag{7}$$

P is the pressure of all the phases. $\overline{\overline{\tau_f}}$ and $\overline{\overline{\tau_s}}$ are the stress tensors of the fluid and solid, respectively. $\vec{g}$ is the acceleration of gravity. $\beta$ is the Gidaspow drag force coefficient and is suitable for various volume fractions of the solid phase, which can be written as follows (Gidaspow) [28].

$$\beta = 150\frac{\alpha_s(1 - \alpha_f)\mu_f}{\alpha_f d_s^2} + 1.75\frac{\alpha_s]\rho_f\left|\vec{v_s} - \vec{v_f}\right|}{d_s}, \alpha_s \geq 0.2 \tag{8}$$

$$\beta = 0.75C_D\frac{\alpha_s \alpha_f \rho_f\left|\vec{v_s} - \vec{v_f}\right|}{d_s}\alpha_f^{-2.65}, \alpha_s \leq 0.2 \tag{9}$$

The $d_s$ is the diameter of the particle. $C_d$ is the drag coefficient.

The motion equation of particle phase in the DDPM model is not solved in Equation (5). It is computed by Tong [30] and Gidaspow [28], as follows:

$$\frac{d\vec{v_s}}{dt} = \frac{\vec{g}\left(\rho_s - \rho_f\right)}{\rho_s} + F_D(\vec{v_f} - \vec{v_s}) + \vec{F_{KTGF}} \tag{10}$$

$$F_D = \frac{18\mu_f}{\rho_s d_s^2}\frac{C_D Re_s}{24} \tag{11}$$

$$\vec{F_{KTFG}} = -\frac{1}{\alpha_s\rho_s}\nabla\cdot\overline{\overline{\tau_s}} \tag{12}$$

The $Re_s$ and $C_d$ are expressed by (Gidaspow) [28]

$$Re_s = \frac{d_s\rho_f\left|\vec{v_s} - \vec{v_f}\right|}{\mu_f} \tag{13}$$

$$C_D = \frac{24}{\alpha_f Re_s}\left[1 + 0.15(\alpha_f Re_s)^{0.687}\right] \tag{14}$$

where $Re_s$ is the Reynolds number of solid.

In the kinetic theory of granular flow (KTGF), the shear viscosity of solid ($\mu_s$) contains kinetic viscosity ($\mu_{s,kin}$), collisional viscosity ($\mu_{s,col}$), and fractional viscosity ($\mu_{s,fri}$).

$$\mu_s = \mu_{s,kin} + \mu_{s,col} + \mu_{s,fri} \tag{15}$$

Kinetic viscosity can be expressed as [31]

$$\mu_{s,kin} = \frac{10d_s\rho_s\sqrt{\Theta_s\pi}}{96\alpha_s g_0(1+e)}\left[1 + \frac{4}{5}\alpha_s g_0(1+e)\right]^2 \tag{16}$$

$\Theta_S$ is the granular temperature. The $g_0$ is the radial distribution function, which represents the resistance of particles to deformation and can be obtained by [32]

$$g_0 = \left(1 - \left(\frac{\alpha_s}{\alpha_{s,max}}\right)^{\frac{1}{3}}\right)^{-1} \tag{17}$$

The collisional viscosity can be written as [30]

$$\mu_{s,col} = \frac{4}{5}\alpha_s d_s\rho_s g_0(1+e)\left(\frac{\Theta_s}{\pi}\right)^{\frac{1}{2}} \tag{18}$$

The fractional viscosity can be written as [33]

$$\mu_{s,fri} = \frac{P_{friction}Sin\Phi}{2\sqrt{I_{2D}}} \tag{19}$$

$$P_{friction} = 0.1\alpha_s\frac{(\alpha_s - \alpha_{s,min})^2}{(\alpha_{s,max} - \alpha_s)^3} \tag{20}$$

$$\lambda_s = \frac{4}{3}\alpha_s d_s\rho_s g_0(1+e)\left(\frac{\Theta_s}{\pi}\right)^{\frac{1}{2}} \tag{21}$$

where the $\varphi$ is the angle of flow direction.

The pressure of the solid phase can be expressed as [34]

$$P_s = \alpha_s \rho_s \Theta_s + 2\alpha_s^2 \rho_s \Theta_s g_0 (1 + e) \tag{22}$$

### 2.3. Rough and Tortuous Model Development

An outcrop sample is cut into cubes with side lengths of 300 mm, which is shown in Figure 3. Then, rough fracture surfaces are obtained through the tri-axial hydraulic fracturing with the mimicked reservoir condition. The rough fracture surfaces are 3D scanned to calculate their Ra. Then, this pair of fracture surfaces is duplicated through 3D printing and installed into the transparent fractures flow model, and 20/40 mesh sands with a volume fraction of 7.4% are injected into the model for observing the formation and transport of the sand bank. A 3D fracture model is built and the dimension of the fracture is consistent with our experimental fracture-flow model. The total generated mesh number of the fracture is 151,392, as shown in Figure 4, and all the mesh meets the calculation requirements. The mesh quality is represented on the x-axis, with the x values closer to 1 indicating better mesh quality. The y-axis represents the number of grid cells, and the upward blue arrow in the figure indicates that the actual number of meshes is much larger than the upper limit of the coordinate. The experimental results are further compared with simulation results using our simulation model.

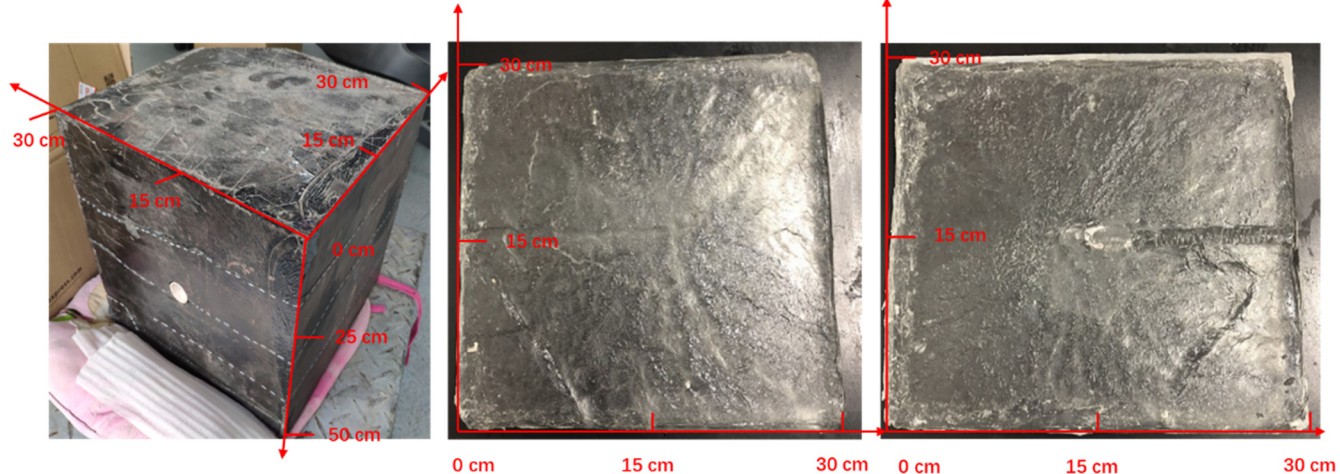

**Figure 3.** Outcrop sample and duplicated fractures.

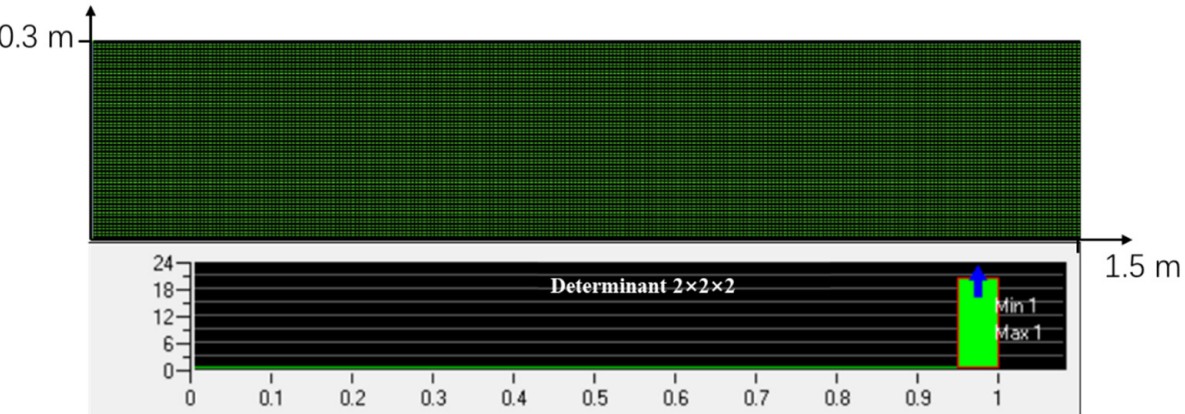

**Figure 4.** Generated mesh and mesh quality.

The roughness of a fracture surface can be quantified in different ways. Babadagli [21] used variogram ($D_{va}$), power spectral density ($D_{psd}$), and triangular prism analyses ($D_{tp}$) to describe the roughness of a fracture. In this study, the integral of surface heights is used,

as defined in Equation (23). In this study, conglomerate and shale fractures are used, which have *Ra* values of 0.39 and 0.74, respectively, as shown in Figure 5.

$$R_a = \frac{1}{l} \int_0^l |y(x)| dx \tag{23}$$

*l* is the sampling length and $y(x)$ is the height of points on the chosen line.

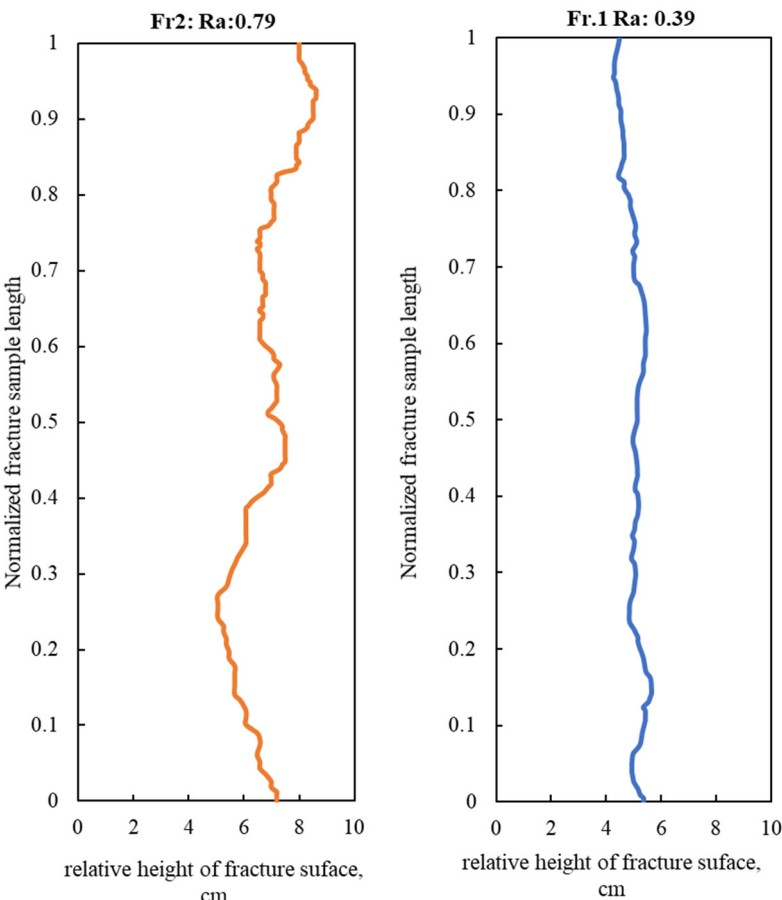

**Figure 5.** Roughness of fractures with different Ra.

In ANSYS Fluent, the roughness of a fracture surface is defined by Ks. As shown in Figure 6, it is assumed that the rough fracture surface is formed by a layer of closely packed balls with an identical radius, and the radius of the ball is Ks. When Ks equals to zero, a smooth fracture surface is obtained.

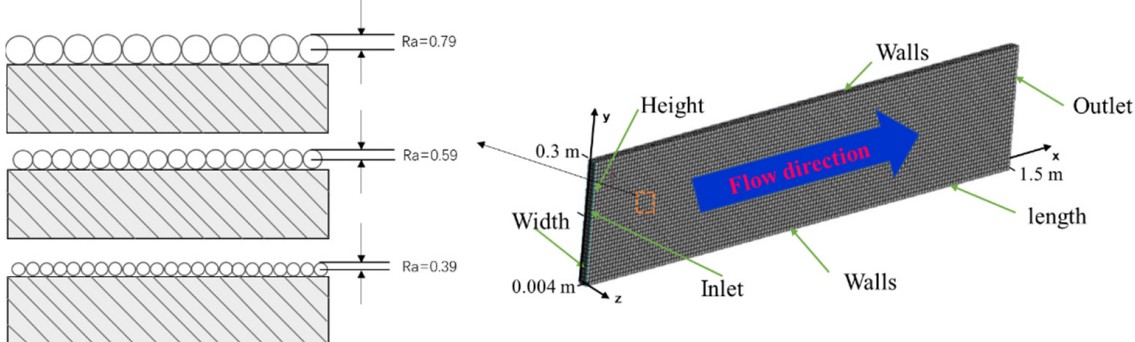

**Figure 6.** Straight fracture roughness model.

Besides the straight fracture, a fracture with a curvature is used to study the sand transport in fractures with near wellbore tortuosity, as is shown in Figure 7. In this tortuous fracture model, the angle of the curvature is 120° and two kinds of curved fractures are used, which are 0.2 m and 0.75 m from the velocity-inlet boundary, respectively. Three sensitive parameters are studied including the slurry velocity, volume fraction of sands, and sands diameter. The parameters in the simulation process are shown in Table 4.

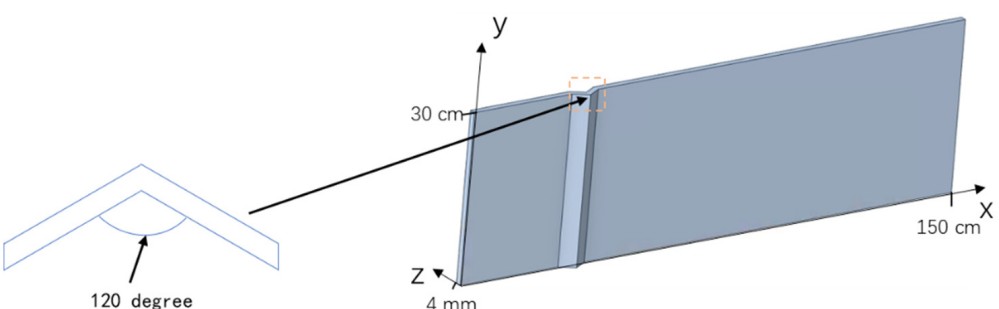

**Figure 7.** Tortuous fracture model.

**Table 4.** Parameters of simulation model.

|  | Straight Facture | Tortuous Facture 1 | Tortuous Facture 2 |
|---|---|---|---|
| length, m | 1.5 | 1.5 | 1.5 |
| width, m | 0.004 | 0.004 | 0.004 |
| height, m | 0.3 | 0.3 | 0.3 |
| tortuous angle, degree |  | 120 | 120 |
| tortuous length, m |  | 0.06 | 0.06 |
| tortuous position, m |  | 0.75 | 0.2 |
| tortuous width, m |  | 0.00346 | 0.00346 |

## 3. Results and Discussion

### 3.1. Validation of Numerical Model

Figure 8 shows the results of the comparison between the simulation results and experiment results. Water and 40/70 mesh sands are injected at the inlet at different velocities, which are 12 m/min, 24 m/min, and 36 m/min.

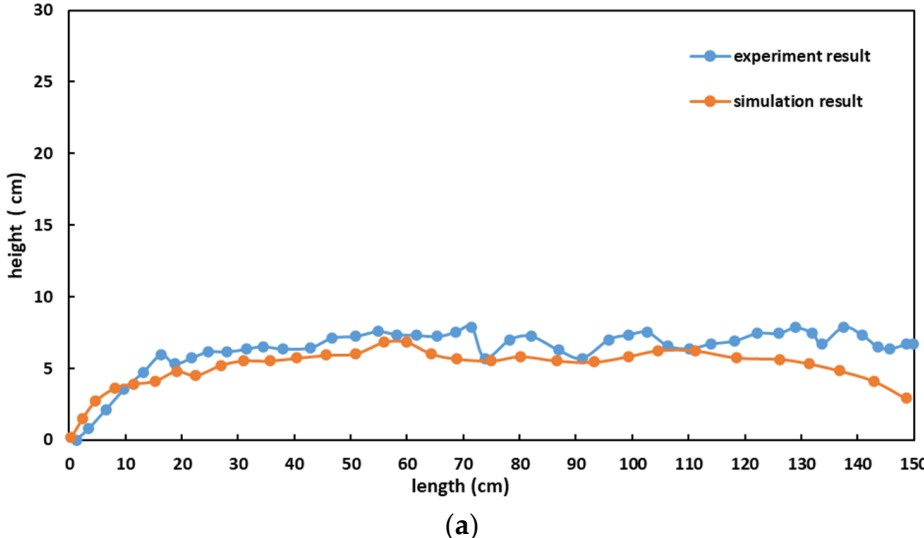

(**a**)

**Figure 8.** *Cont*.

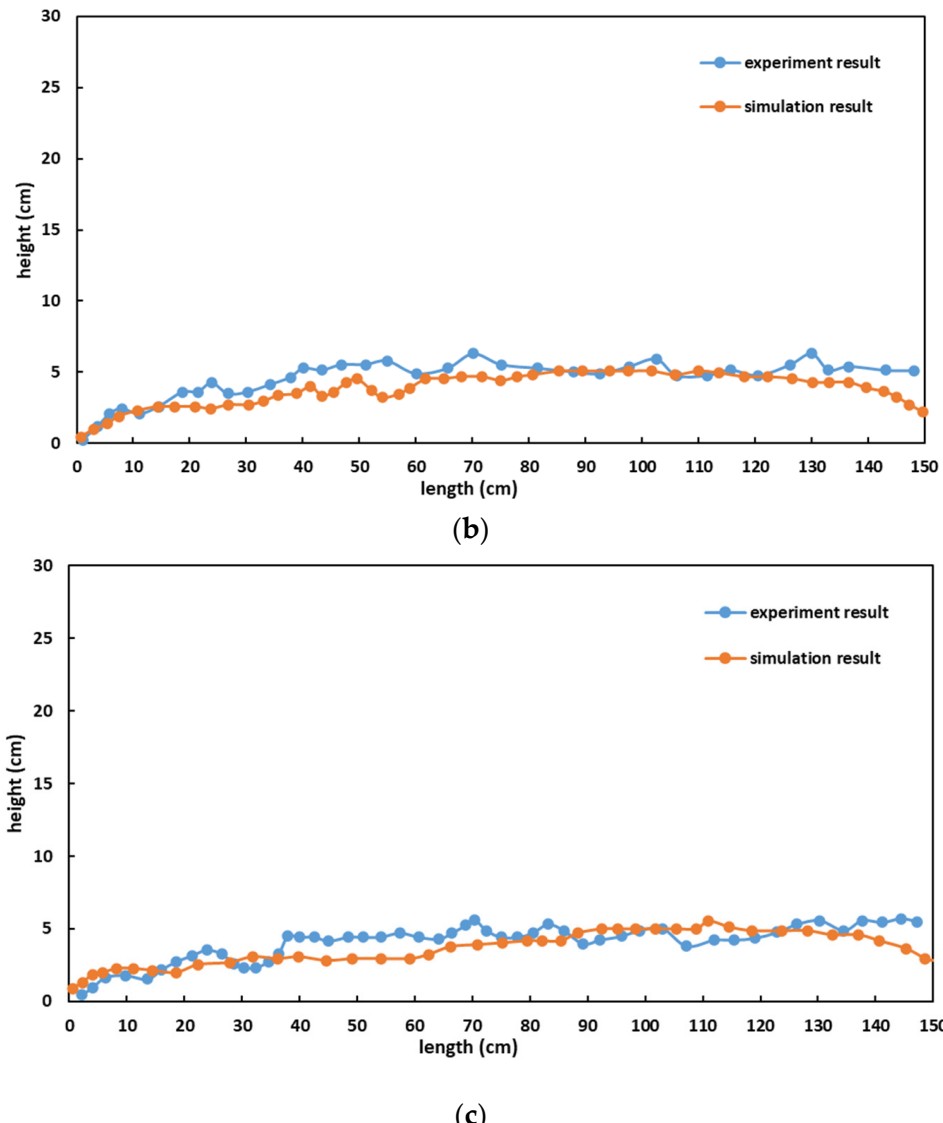

**Figure 8.** The results comparison between experiments and simulation at t = 50 s. (**a**) v = 12 m/min, (**b**) v = 24 m/min, and (**c**) v = 36 m/min.

When the injection rate is 12 m/min, the equilibrium height of the sand bank is 5.75 cm and 5.42 cm for the experiment and simulation results. The comparison between experiments and simulation results had an average percentage error of 3.34% for the equilibrium height of the sand bank. This suggests a well consistency between experiments and simulation results.

### 3.2. Sand Transport in Rough Fractures
3.2.1. Effect of Slurry Velocity

The sensitivity analysis in this section involves three key factors including slurry velocity, sand diameter, and sand volume fraction. At first, the slurry velocity is changed, keeping all other factors constant.

Figure 9 shows the change in sand distribution in different cases when the slurry injection velocity is 12 m/min, 24 m/min, and 36 m/min. When the injection velocity is 12 m/min, the equilibrium height of the sand bank is 13.00 cm. In the region where the sand bank is stabilized, the sand volume fraction is around 0.59—0.63, which is the density of the random packing of spheres. In the region where sands are floating in the fracturing fluid, the volume fraction is 7.3%, which is identical to that of the injected slurry. When the

sand bank is formed and the equilibrium height is reached, the farthest horizontal distance reached after the complete settlement of sands can be found, as pointed by a black arrow in Figure 10. This horizontal distance is 3.63 m when the slurry injection rate is 96 m/min, and it decreases to 0.46 m when the slurry injection velocity is 12 m/min (Figure 11).

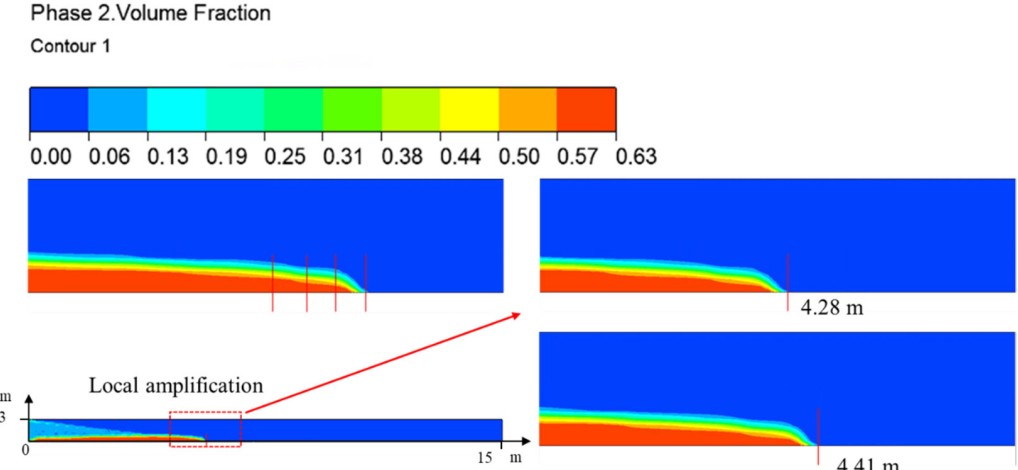

**Figure 9.** Acquisition of propagation rate of the propped fractures in the smooth fracture.

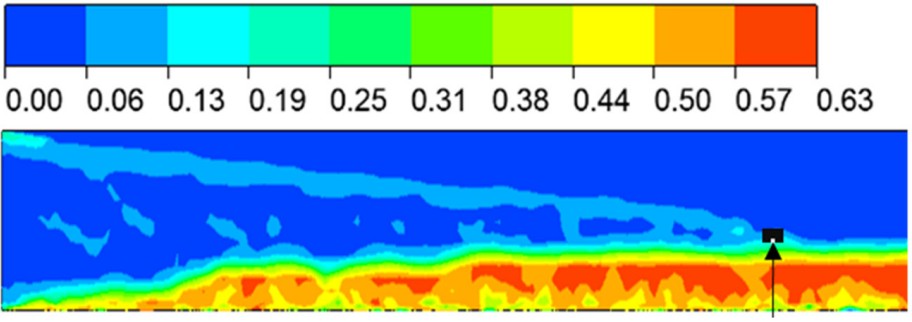

**Figure 10.** Sand distribution at the end of simulation (black arrow points to the farthest location of sand settlement).

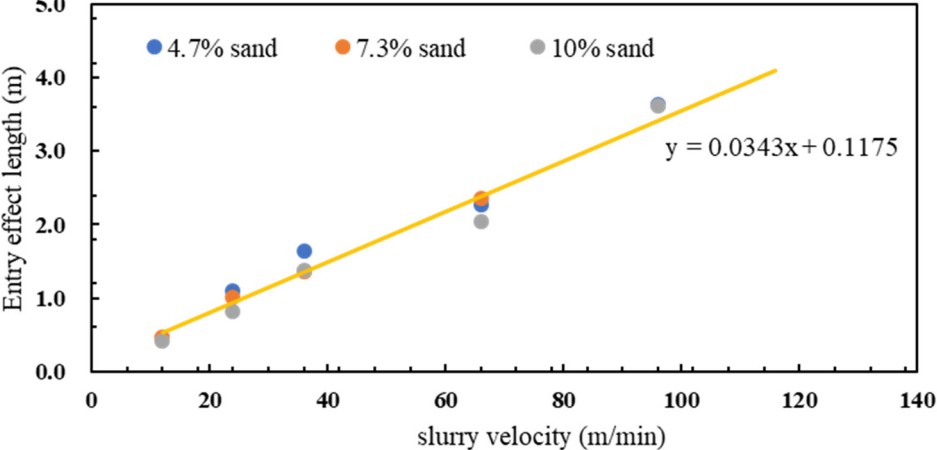

**Figure 11.** Entry effect length with slurry horizontal velocity.

From the simulation, the locations of the sand bank front can be marked at different time slices, from which the transport rate of the sand bank can be calculated. Figure 9 shows a schematic diagram to calculate the propagation rate of the propped fractures. As is shown in Figure 12, it increases with the injection rate and the volume fraction of sand injected.

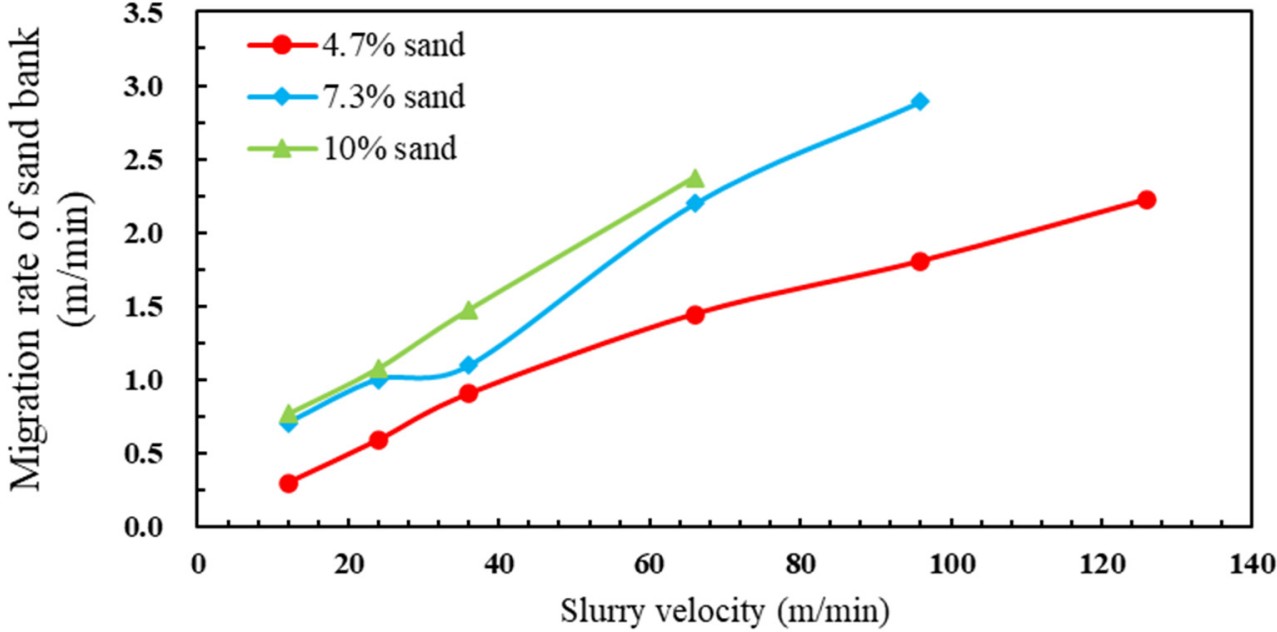

**Figure 12.** Changes in propagation rate of the propped fractures with slurry injection rate and sand volume fraction.

The simulation results of sand volume fraction contour plots are shown in Figure 13. The three cases of variation in the slurry injection velocity are 12 m/min, 24 m/min, and 36 m/min, respectively. On comparison of the sand volume fraction contour plot for different slurry injection velocity, it can be found that as the velocity of slurry increases, the sand bank height decreases and the propped length of fractures increases. Figure 12 shows the propped length and propped height of fractures in the simulation results of different slurry velocity. As the slurry velocity increases, the propped height decreases and the propped length increases. A new parameter entry effect length shown in Figure 10 is put forward to explain this phenomenon, which represents the sand settling position at the farthest position of the sand bank. This is to say, after the sand bank reaches equilibrium height, the sand can be carried by the fluid to the farthest position on the sand bank under the influence of the horizontal fluid flow. Additionally, the distance between the farthest position of the sand settlement and fracture inlet is the entry effect length. It describes the degree to which the sand bank at the entrance is washed out by the subsequent slurry after equilibrium [10]. As is shown in Figure 11, the entry effect length and horizontal slurry velocity have a linear relationship, which suggests that increasing the horizontal slurry velocity has no effect on the sand settling velocity. Since the settling velocity of the particles is not disturbed by the horizontal slurry velocity, at a certain same time, more sand will be transported away from the inlet boundary. Moreover, the high slurry injection velocity strengthens the force between fluid phase and solid phase, resulting in more sand carried deeper into the fractures leading to a shorter sand bank.

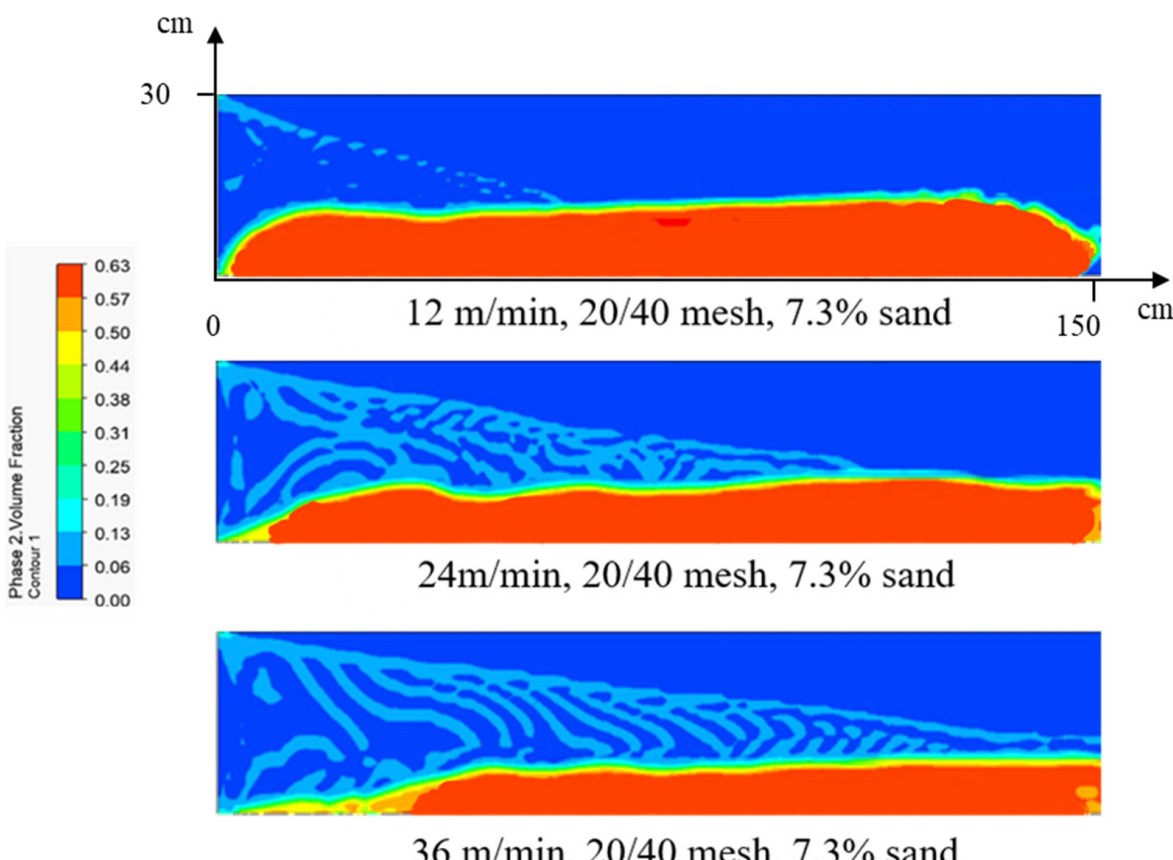

**Figure 13.** Comparison of sand distribution for different slurry velocities.

The present study of the sand distribution to injection velocity suggests that it is of great importance in optimizing sand transport and distribution. When the slurry velocity is doubled, the propped length of the fracture will basically double, so the drainage area of the reservoir fracture is greatly improved, and the conductivity of the fracture tip is also improved. Especially for the low viscosity fracturing fluid, such as slick water fracturing, a higher injection velocity will be a good approach to improve the fracturing results.

### 3.2.2. Effect of Sand Volume Fraction

After the investigation of the effect of slurry velocity, the effect of the sand volume fraction is studied. Figure 14 shows the change in sand distribution in different cases when the sand injection is 4.7%, 7.3%, and 10.0%. When the sand injection is 7.3%, the length of the sand bank is 0.85 m and increases to 0.98 m when the sand injection is 10.0%. The sand bank length increases with the volume fraction of sand injection as shown in Figure 11.

Improving the volume fraction of sand injection can effectively reduce the time for the sand bank to reach the equilibrium height and accelerates the rate of the sand bank moving forward. In other words, in the same fracturing time, if the volume fraction of injected sand is increased, the longer of the fractures would be propped. Therefore, in order to reduce field working time, increasing the volume fraction of sand injection is a practical approach. During the actual production process, the hydraulic fractures are not always straight, but tortuous and nonuniform. Thus, using a high-volume fraction of sand can cause sand plug, especially when using low viscosity fracturing fluid such as slick water. In our study, the sand bank is studied without generation sand-plugging or screening-out. A reasonable volume fraction of sand injected will be a good choice that can decrease the hydraulic fracturing time and avoid sand plug.

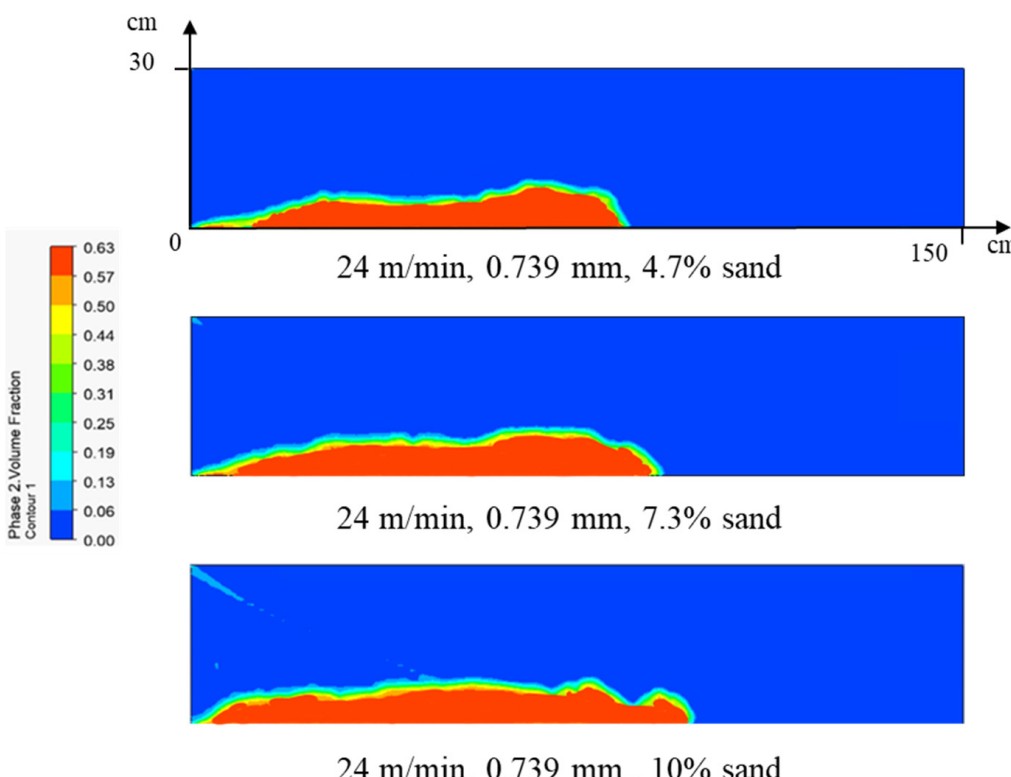

**Figure 14.** Comparison of sand distribution for different sand volume fractions.

### 3.2.3. Effect of Sand Diameter

Next, the effect of the sand diameter on the sand bank equilibrium height and the propped length is studied. Figure 15 shows the change in sand distribution in different cases when the sand diameters are 0.208 mm, 0.325 mm, 0.425 mm, and 0.739 mm. When the sand diameter is 0.739 mm, the equilibrium height of the sand bank is 13.00 cm and it decreases to 2.85 cm when the sand diameter is 0.285 mm. Figure 16 shows the relationship between the sand bank height, sand diameter, and slurry velocity. It can be found that as the diameter decreases, the sand bank equilibrium height apparently decreases, which can be observed in the left of Figures 17 and 18. The rate of propped length is not affected by the sand diameter, which can be observed in the right column of Figure 17.

To understand the reason for this phenomenon, a graph is plotted against variables entry effect length/slurry velocity and sand diameter, as shown in Figure 19, where entry effect length is clarified earlier. When the horizontal flow rate changes, the entry effect length/slurry velocity of the same sand size has the same value, indicating that the horizontal flow has little effect on the settling of proppant. It can be interpreted that as the sand diameter increases, the ratio of entry effect length/slurry velocity decreases, which suggests that a large diameter of sand tends to settle at the fracture entrance. Thus, it can be reasonable to explain the formation process of sand bank. In the case of the same horizontal injection flow rate, at the same time, due to the large sedimentation velocity of the sand of large particle size, most of the sand settled, while the small sized sand can be transported away from the wellbore. Thus, at the beginning of sand formation, large sized particles show a high sand bank. Moreover, sands with large particle sizes have large mass and large inertia. When the inlet flow rate is constant, the distance between the sand bank and the top of fracture is smaller, and the velocity will be larger. So, small sized particles can be transported easily by a low velocity of fracturing fluid while large sized particles require a high velocity of fracturing fluid to wash away. Therefore, the sand bank height of different particle diameters shows differently.

The sedimentation speed of large particles is fast, and the sand bank is high. The sedimentation speed of small particles is slow, and the sand bank is short, so the propped rate of sands of large and small particles in fractures is relatively close.

With the increase in sand particle size, the height ratio of propped fractures becomes higher, and the migration rate of the sand bank does not change much with the particle size. In the field scale, the small-sized proppant can be pumped first, so that the distal fracture can be effectively supported, and then the slightly larger-sized proppant can be pumped in. The overall fracture is effectively supported, and the propped length of the fracture is further improved to improve the conductivity of the fracture.

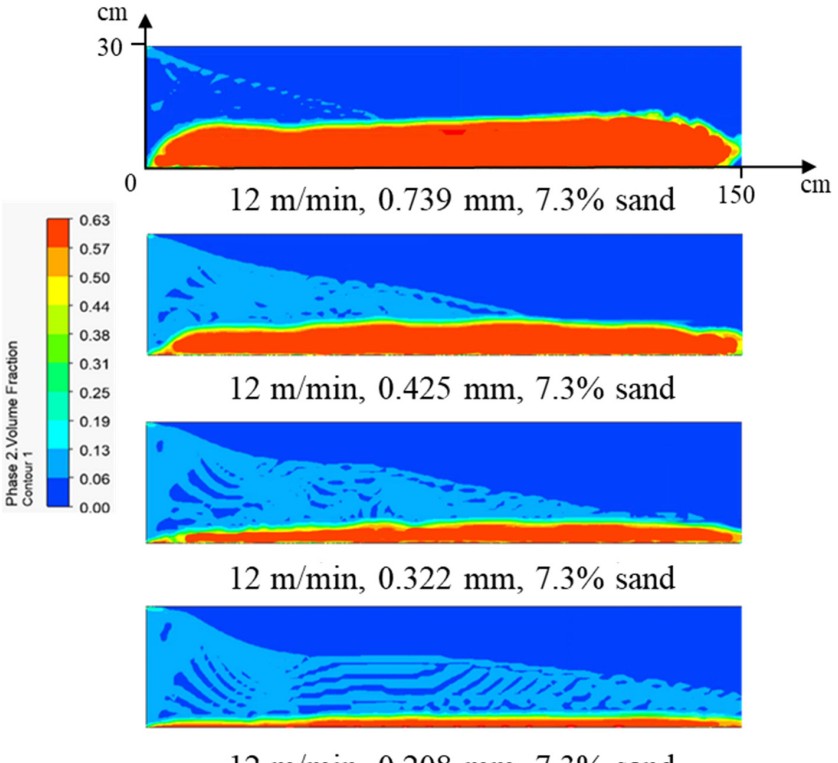

**Figure 15.** Comparison of sand distribution for different sand diameters.

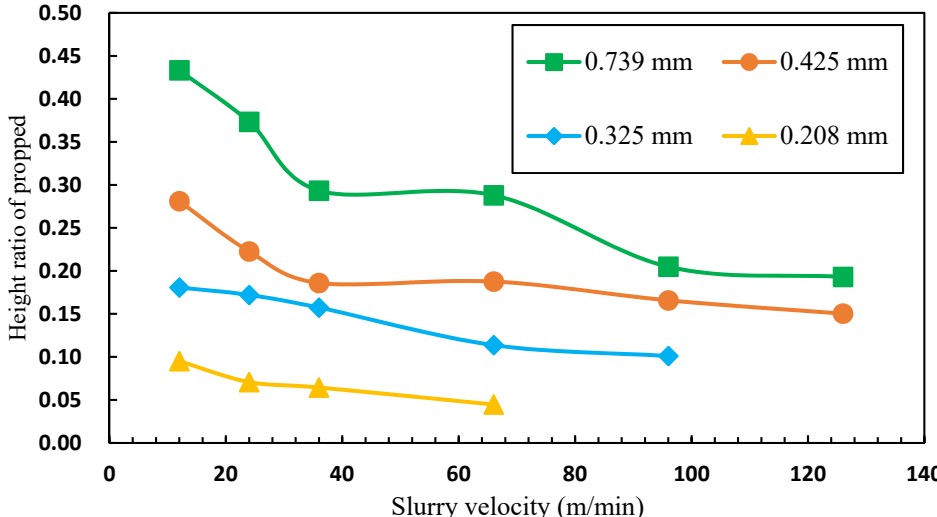

**Figure 16.** Changes in sand bank equilibrium height with slurry velocity and sand diameter. (The y-axis represents the percentage of sand bank height in the fracture.).

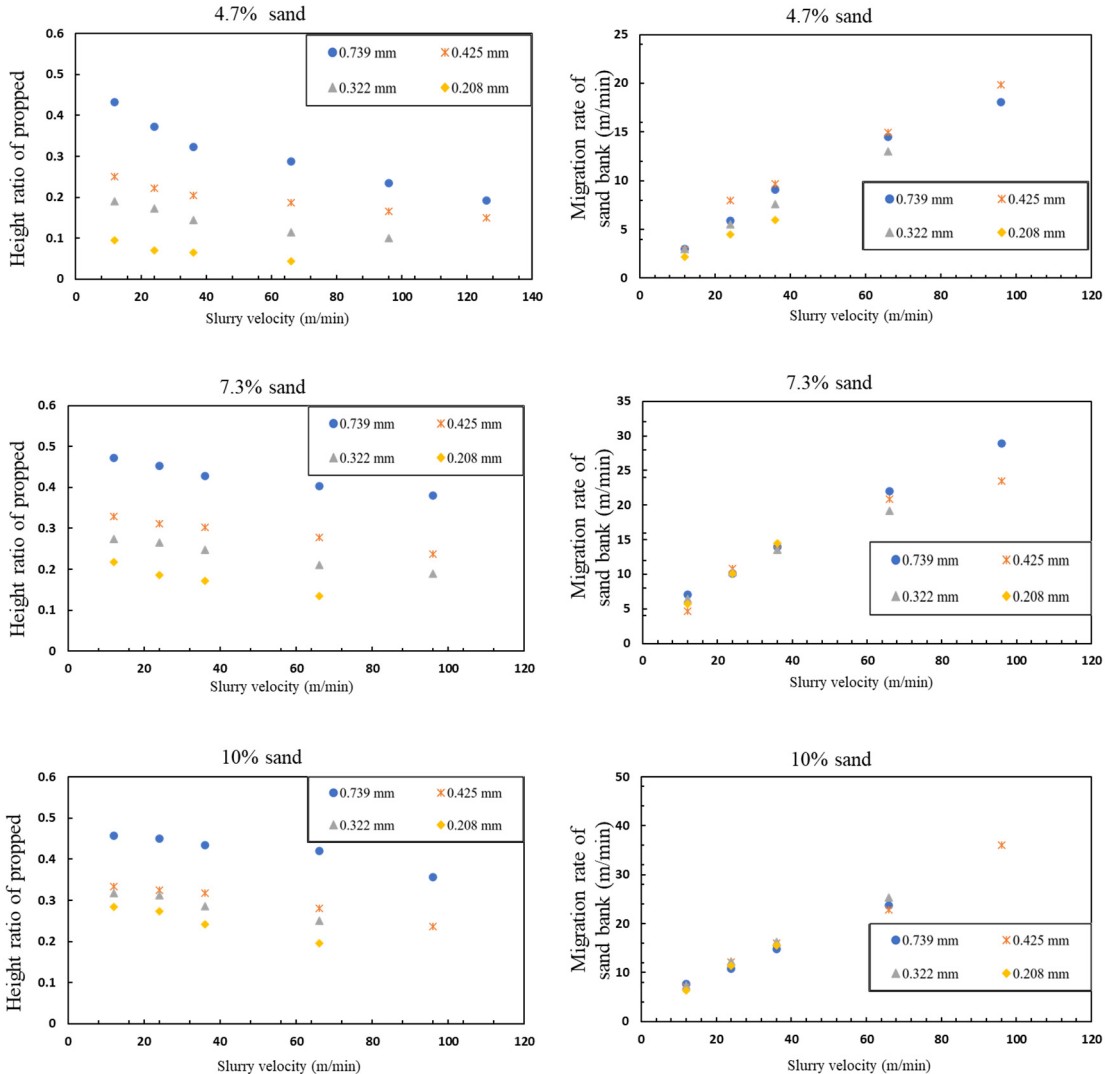

**Figure 17.** Changes in sand bank equilibrium height ratio (left column; the y-axis represents the percentage of sand bank height in the fracture) and propagation rate of propped fractures (right column) with slurry velocity, sand diameter, and sand volume fraction.

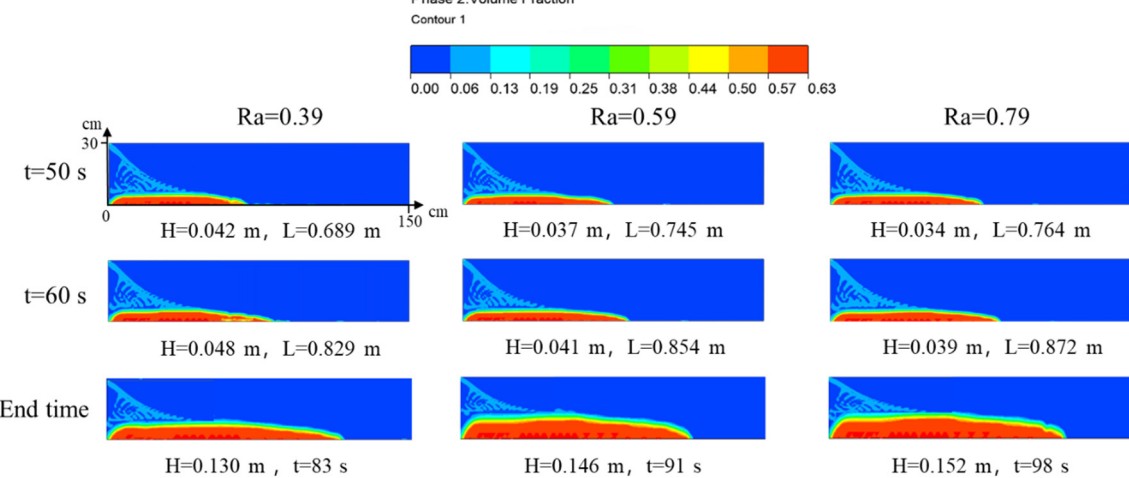

**Figure 18.** Changes in sand bank equilibrium height and propagation rate of propped fractures with Ra (roughness of fractures, slurry velocity 12 m/min, and 20/40 mesh sand; 7.3% sand.).

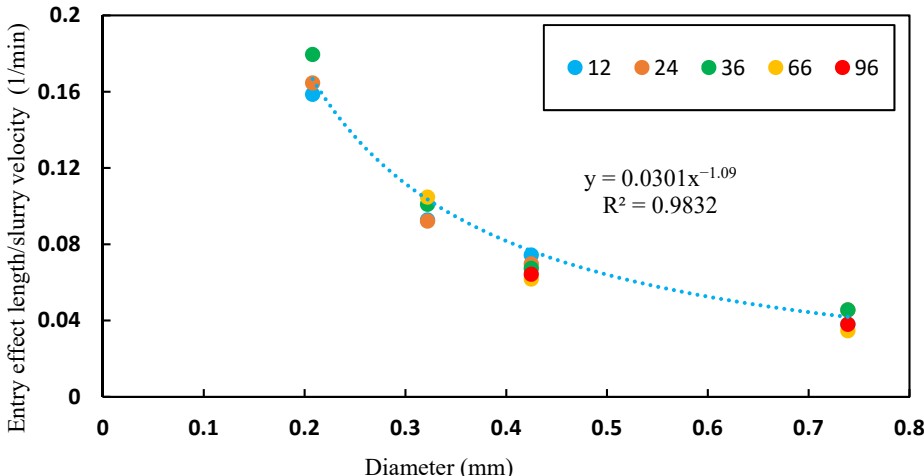

**Figure 19.** Settling velocity with different diameters (the legend means the slurry velocity, and the unit is m/min).

### 3.2.4. Effect of Surface Roughness

Finally, the effect of fracture surface roughness on the transport and distribution of sand is studied. Three types of surface roughness are used for the investigation. Figure 18 shows the change in sand distribution in different cases when the Ras are 0.39, 0.59, and 0.79. When the Ra is 0.39, the equilibrium height of the sand bank is 0.130 m and increases to 0.152 m when Ra is 0.79. As is shown in Figure 20, at the beginning of sand injection, the roughness has no obvious influence on the transport and distribution of the sand. As time goes by, when the sand bank builds up, it can be observed that with the increase in roughness, the height of the sand bank gradually increases, and the moving forward rate of the sand bank in the fractures gradually decreases. It can be explained that as the roughness increases, the force between the particles and the wall is further strengthened. It can be inferred that with the increase in surface roughness, the probability of particle and wall collision is higher, which results in a gradual decrease in particle settling velocity, so the sand can be carried to the fractures by the fracturing fluid. Furthermore, as the fracture roughness increases, at the beginning of sand injection, the sand can be transported further and forms a longer sand bank. At the same time, as the roughness increases, the frictional resistance between the particles and the fracture wall will be stronger. Thus, the distance between the fractures and the height of sand bank will decrease, so that the sand transported with the fracturing fluid and the accumulation of the sand bank reform a new balance. That is to say, a higher sand bank will be formed.

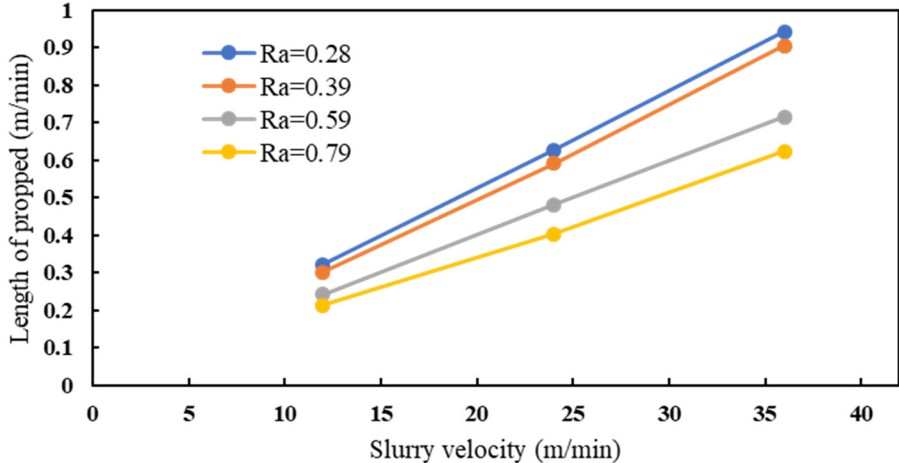

**Figure 20.** Comparison of propped length with different Ra.

### 3.2.5. Effect of the Tortuous Fractures

Next, an analysis is carried out to investigate the impact of near wellbore tortuous fractures on sand transportation and distribution. A comparation is made between straight fractures and tortuous fractures. In this model, the tortuous model is simplified as a 'v' shape model at different places in the fractures. Figure 21 shows the results of sand volume fraction contour plot of different fractures. The contour plot looks the same for the tortuous fractures away from wellbore and straight fractures. As for the near wellbore tortuous fracture, the sand bank is significantly different. In this kind of fracture, the sand tends to form a plug and the resulting later slurry is difficult to inject.

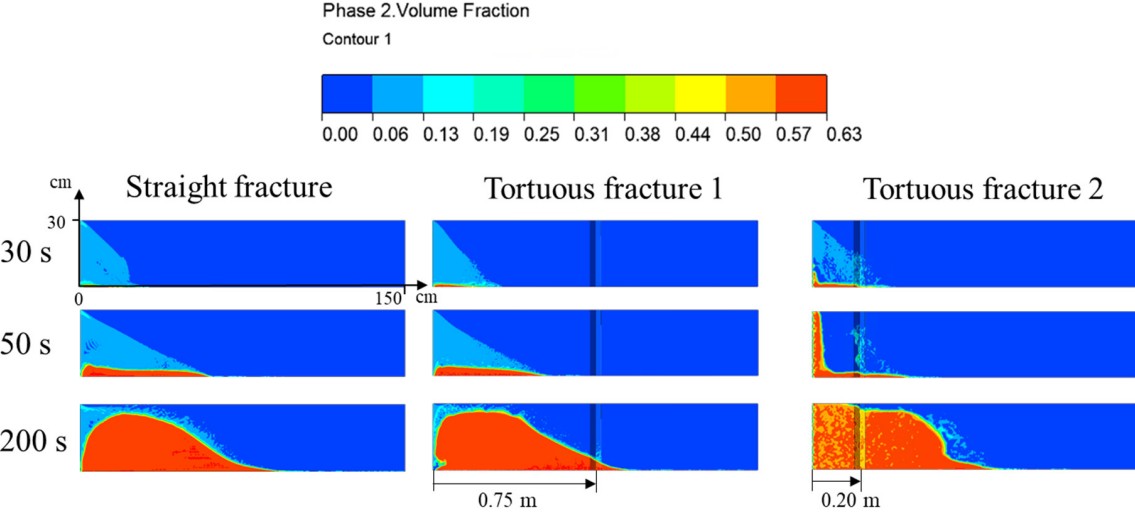

**Figure 21.** Sand volume fraction contour plot with different positions of tortuous fractures.

A velocity vector plot is put forward to explain this situation. As shown in Figure 22, in the near wellbore tortuous model, the horizontal velocity of the granular phase apparently decreases. The appearance of near wellbore tortuous fractures creates an additional resistance to sand to transport. When the sand bank reaches equilibrium at the position, due to the unsteady flow conditions at the tortuous position, it is easy to produce vortices. The generation of vortices will cause the accumulated sand to be transported again. Thus, there is a significant height of sand bank change at the tortuous position.

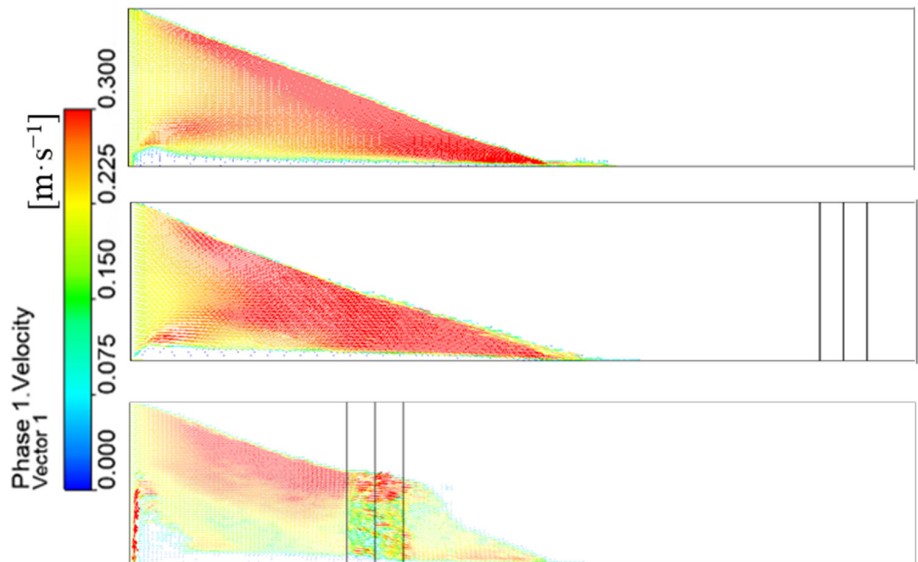

**Figure 22.** Velocity vector analysis in different fractures.

## 4. Conclusions

In this study, sand transport and distribution in rough and tortuous fractures is investigated using the Dense Discrete Phase Model (DDPM). The surface height integral (Ra) is quantitatively used to study the effect of fracture roughness on sand movement. Three sensitive parameters are analyzed to study the key factors that influence the sand transport and distribution including injection rate, sand diameter, sand volume fraction, and roughness of fractures.

Among all the factors, slurry velocity is the most important parameter affecting the sand transport and distribution. In order to reduce field working time, increasing the volume fraction of sand injection is a practical approach. Compared with smooth fractures, rough fracture walls employ resistance during sand transport, reducing the settling velocity of sand in fractures. With the increase in Ra, the sand bank equilibrium height increases and the rate of propped length decreases. Based on the simulation results, an analytical solution is built that can easily predict field scale sand transport and distribution. Sand can effectively migrate to the fracture's tip and prop the fracture quickly by increasing the pumping flow rate in field. As for the rough fractures in the field, the propped fracture would be longer by reducing the sand size and increasing the volume fraction of sand.

In the tortuous fractures model, the flow phenomenon is different from that in straight fractures. Due to the 'v' shaped model, the slurry velocity obviously decreases and leads to a higher sand bank and slower propped length. In field construction, it is expected that the fractures can be propped evenly. However, most of the fractures formed are often curved near the wellbore. Through experiments and simulation, it is found that reducing the sand size can effectively reduce the sand migration resistance caused by this near-wellbore tortuosity.

**Author Contributions:** Methodology, D.W. (Di Wang) and B.B.; software, B.B.; validation, B.W. and D.W. (Dongya Wei); investigation, B.B. and T.L.; data curation, B.B. and D.W. (Dongya Wei); writing—original draft preparation, D.W. (Di Wang) and B.B.; writing—review and editing, B.W., D.W. (Dongya Wei), and T.L.; supervision, T.L.; funding acquisition, D.W. (Di Wang) and T.L. All authors have read and agreed to the published version of the manuscript.

**Funding:** This work was financially supported by the Strategic Cooperation Technology Projects of CNPC and CUPB (ZLZX2020-01) and by the National Energy Shale Oil Research and Development Center (33550000-20-FW2099-0156).

**Institutional Review Board Statement:** Not applicable.

**Informed Consent Statement:** Not applicable.

**Data Availability Statement:** The data presented in this study are available on request from the corresponding author.

**Conflicts of Interest:** The authors declare no conflict of interest.

## Nomenclature

| | |
|---|---|
| $\overline{\overline{\tau_f}}$ | the stress tensors of the fluid |
| $\overline{\overline{\tau_s}}$ | the stress tensors of the solid |
| $\overrightarrow{g}$ | the acceleration of gravity |
| $v_f$ | velocity of fluid |
| $v_s$ | velocity of solid |
| $\Theta_S$ | granular temperature |
| $\alpha_f$ | volume fraction of fluid |
| $\alpha_s$ | volume fraction of solid |
| $\rho_f$ | density of the fluid |
| $\rho_s$ | density of the solid |
| Cd | the drag coefficient |

| | |
|---|---|
| d | hydraulic diameter |
| ds | the diameter of the particle |
| g0 | radial distribution function |
| P | pressure of all the phases |
| Re | Reynolds number |
| Res | Reynolds number of solid |
| v | velocity |
| β | the Gidaspow drag force coefficient |
| μ | viscosity |
| μs,col | collisional viscosity |
| μs,fri | fractional viscosity |
| μs,kin | kinetic viscosity |
| μs | shear viscosity of solid |
| ρ | density |

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
