# Peer review of "Impacts of Fracture Roughness and Near-Wellbore Tortuosity on Proppant Transport within Hydraulic Fractures"

_sustainability, doi:10.3390/su14148589_

Round 1

Reviewer 1 Report

This is a very useful numerical modeling experiment with some useful observations. However, in my view, some improvements are necessary before it can be considered for publication. some of my comments are as follows:

1. There are significant language issues throughout the manuscript. This includes both spelling and grammatical issues as well as structure of many paragraphs/ sentences. While it is impossible for me to tabulate all of the issues, I would suggests making use of professional english editing service to re-draft the manuscript.

2. When looking at the background from field test sites, I would suggest authors look at roughness vs. proppant concentration studies done at HFTS-1 site. Noteworthy is https://doi.org/10.2118/204174-MS. The paper demonstrates the close relationship between roughness and proppant concentration in hydraulic fractures. It is worthwhile to reference such studies to highlight actual field observations to demonstrate impact of rough fractures.

3. An important factor to consider is this bed boundaries. Fractures crossing across such boundaries tend to show significant complexity creation which impacts proppant entrapment, sand bank build-up. This should also be discussed in the background discussion.

4.   It is unclear how the lab scale data is upscaled to field scale for comparison. If that was not required, it will be beneficial to provide some discussion as to why/ why not.

5. Page 3, line 145: Is there a reason why the figure accompanying the discussion is shared later (Figure 8)? It is ideal if the figures being referred to are proximal to the discussion.

6. Is a simplistic roughness formulation such as Ks as discussed in Page 8 relatable to complex roughness characteristics such as morphology changes observed in real rock samples as shown in Figure 3 or observed in recent core-through experiments? If so, how is it justified?

7. Figure 4: I am not sure if this is required.

8. Page 9, line 281: the angle of curvature is mentioned as 120 degrees. However, it is a bit unclear. w.r.t what axis? It might be worthwhile to update Figure 7 to highlight relevant parameters such as curvature angle.

9. Figure 8: all test cases show divergence at larger lengths. Will it be useful to test the convergence at larger lengths  (> 150 cm)? It is possible that at length, the governing factors might be significantly different from assumptions used in the simulation.

10. Page 13, Line 350: authors talk about using higher injection velocity. How about the interplay between slurry velocity and bottom-hole fluid pressure? Normally, in many formations, the rates are limited by pressure considerations.

11. Page 15, Line 374 onwards: What about issues with premature screenouts which is the case in most plays in US? With higher probability of screenouts indicating costly clean-up operations and lost frac-time. The authors do mention this as an issue but most operators are already operating close to the limits as is.

12. While the authors have shown the sand back profiles, it might have been useful to also map our pressure variations (for example, at injection or crack tip) for reference and better analysis of some of the interpretations authors have made.

Author Response

Manuscript Submission ID sustainability-1760450

Response to Reviewers

Dear Reviewers,

We sincerely appreciate your time and effort on your careful reading this manuscript, understanding all the details, as well as providing valuable comments. These comments were very helpful and we believe have greatly improved the quality of this manuscript.

For ease of the reviewers, we repeat each comment, give a short reply, and detail our changes made to each comment in this document.

Again, thanks for your consideration.

Sincerely,

Di Wang

Bingyang Bai

Bin Wang

Dongya Wei

Tianbo Liang

 Editor comments:

This is a very useful numerical modeling experiment with some useful observations. However, in my view, some improvements are necessary before it can be considered for publication. some of my comments are as follows:

Reviewer Comments:

Reviewer 1

  1. There are significant language issues throughout the manuscript. This includes both spelling and grammatical issues as well as structure of many paragraphs/ sentences. While it is impossible for me to tabulate all of the issues, I would suggests making use of professional English editing service to re-draft the manuscript.
  • The language throughout the manuscript has been modified.
  1. When looking at the background from field test sites, I would suggest authors look at roughness vs. proppant concentration studies done at HFTS-1 site. Noteworthy is https://doi.org/10.2118/204174-MS. The paper demonstrates the close relationship between roughness and proppant concentration in hydraulic fractures. It is worthwhile to reference such studies to highlight actual field observations to demonstrate impact of rough fractures.

Change1: The impact of rough fractures in actual field observations has been added.

Original Version

Numerous experimental and simulation studies are conducted to investigate the transport and distribution of proppant in fractures. The motion of proppant in the fractures can be divided in two aspects. The settling of proppant in the vertical direction and transport in the horizontal direction1. It can be inferred that investigating the law of proppant settling is of great significance2. Kern et al.3 found that the proppant settling velocity decreases with the particle diameter closing to fracture width. Shrivastava and Sharma4 found that the smaller particles have good suspension and can transport deeper in to the natural fractures. Harrington et al.5 put forward a modified stokes formula to describe proppant settling in non-newtonian fluid. Proppant settling velocity will decrease when the concentration of proppant increases when the particles are not clustered6. Gadde et al.7 investigated proppant settling in rough fractures and found that the proppant settling velocity decreased when compared to the smooth facture.

Current Version

Numerous experimental and simulation studies are conducted to investigate the transport and distribution of proppant in fractures. The motion of proppant in the fractures can be divided in two aspects. The settling of proppant in the vertical direction and transport in the horizontal direction1. It can be inferred that investigating the law of proppant settling is of great significance2. Kern et al.3 found that the proppant settling velocity decreases with the particle diameter closing to fracture width. Shrivastava and Sharma4 found that the smaller particles have good suspension and can transport deeper in to the natural fractures. Harrington et al.5 put forward a modified stokes formula to describe proppant settling in non-newtonian fluid. Proppant settling velocity will decrease when the concentration of proppant increases when the particles are not clustered6. Gadde et al.7 investigated proppant settling in rough fractures and found that the proppant settling velocity decreased when compared to the smooth facture. There is a strong positive correlation between proppant placement concentration and local fracture roughness8.

  1. An important factor to consider is this bed boundaries. Fractures crossing across such boundaries tend to show significant complexity creation which impacts proppant entrapment, sand bank build-up. This should also be discussed in the background discussion.

Change1: The impact of bed boundaries and fractures crossing across boundaries are added.

Original Version

Numerous experimental and simulation studies are conducted to investigate the transport and distribution of proppant in fractures. The motion of proppant in the fractures can be divided in two aspects. The settling of proppant in the vertical direction and transport in the horizontal direction1. It can be inferred that investigating the law of proppant settling is of great significance2. Kern et al.3 found that the proppant settling velocity decreases with the particle diameter closing to fracture width. Shrivastava and Sharma4 found that the smaller particles have good suspension and can transport deeper in to the natural fractures. Harrington et al.5 put forward a modified stokes formula to describe proppant settling in non-newtonian fluid. Proppant settling velocity will decrease when the concentration of proppant increases when the particles are not clustered6. Gadde et al.7 investigated proppant settling in rough fractures and found that the proppant settling velocity decreased when compared to the smooth facture.

Current Version

Numerous experimental and simulation studies are conducted to investigate the transport and distribution of proppant in fractures. The motion of proppant in the frac-tures can be divided in two aspects. The settling of proppant in the vertical direction and transport in the horizontal direction1. It can be inferred that investigating the law of prop-pant settling is of great significance2. Kern et al.3 found that the proppant settling velocity decreases with the particle diameter closing to fracture width. Shrivastava and Sharma4 found that the smaller particles have good suspension and can transport deeper in to the natural fractures. Harrington et al.5 put forward a modified stokes formula to describe proppant settling in non-newtonian fluid. Proppant settling velocity will decrease when the concentration of proppant increases when the particles are not clustered6. Gadde et al.7 investigated proppant settling in rough fractures and found that the proppant settling ve-locity decreased when compared to the smooth facture. There is a strong positive correla-tion between proppant placement concentration and local fracture roughness8. Debotyam Maity et al9 found that during the fracture propagation period, the stress at the boundary and the lithology change have a great influence on the proppant distribution.

  1. It is unclear how the lab scale data is upscaled to field scale for comparison. If that was not required, it will be beneficial to provide some discussion as to why/ why not.
  • First, we completed the sand migration experiment at the laboratory scale. And numerical simulation is compared with experiments. Then, we carried out a large number of numerical simulations and established the corresponding field-scale regression prediction model according to the numerical results.
  1. Page 3, line 145: Is there a reason why the figure accompanying the discussion is shared later (Figure 8)? It is ideal if the figures being referred to are proximal to the discussion.
  • It is necessary to mention the equilibrium height here. The Figure 8 is referred. A detailed explanation of Figure 8 is in Section 3.1
  1. Is a simplistic roughness formulation such as Ks as discussed in Page 8 relatable to complex roughness characteristics such as morphology changes observed in real rock samples as shown in Figure 3 or observed in recent core-through experiments? If so, how is it justified?()
  • 3D scans are used to obtain the fracture surfaces features. Then, Ks is used to characterized the roughness of fracture. Then, changing Ks in ANSYS-Fluent is consistent with the Ra of the fracture. And it is found that the formed sand bank is close to the experimental result. After that, we directly changed the Ks embedded in the wall boundary condition to study the effect of wall roughness on sand bank formation and sand transport.
  1. Figure 4: I am not sure if this is required.
  • This figure shows the quantity and quality of the divided meshes, indicating that the divided meshes are reliable for the calculation process.
  1. Page 9, line 281: the angle of curvature is mentioned as 120 degrees. However, it is a bit unclear. w.r.t what axis? It might be worthwhile to update Figure 7 to highlight relevant parameters such as curvature angle.

Change1: The Figure has been changed.

Original Version

Figure 7. Tortuous fracture model.

Current Version

Figure 7. Tortuous fracture model.

  1. Figure 8: all test cases show divergence at larger lengths. Will it be useful to test the convergence at larger lengths  (> 150 cm)? It is possible that at length, the governing factors might be significantly different from assumptions used in the simulation.
  • In Figure 8, the overall shape and height of the sand bank are relatively close. According to the observed phenomenon, it is speculated that the slurry velocity can be pumped in at a constant flow rate in the numerical experiment, and the slurry velocity would fluctuate to a certain extent in the experiment.
  1. Page 13, Line 350: authors talk about using higher injection velocity. How about the interplay between slurry velocity and bottom-hole fluid pressure? Normally, in many formations, the rates are limited by pressure considerations.
  • At present, we have not studied the relationship between bottom hole pressure and slurry velocity. In our follow-up research, we would like to investigate this aspect.
  1. Page 15, Line 374 onwards: What about issues with premature screenouts which is the case in most plays in US? With higher probability of screenouts indicating costly clean-up operations and lost frac-time. The authors do mention this as an issue but most operators are already operating close to the limits as is.
  • In this study, the sand bank formation is studied without generating sand-plugging or screening-out. Improving the volume fraction of sand injection can reduce the time for sand bank to reach the equilibrium height and accelerate the rate of sand bank moving forward.
  1. While the authors have shown the sand back profiles, it might have been useful to also map our pressure variations (for example, at injection or crack tip) for reference and better analysis of some of the interpretations authors have made.
  • In straight fractures, we mainly studied the shape of the sand bank. So the pressure variations contour plots are not shown here. And in our further study, in tortuous fractures, we will study the cloud map of pressure distribution.

Thank you again for your suggestions and is very useful for improving our manuscript.

Reviewer 2 Report

The paper is generally well written and contains valuable results. As a Reviewer, I would like to express my positive opinion Before publication, please follow the points listed below.

1. Please explain all symbols in existed Equations, like Eq1

2. Fig. 3 - PLease add scale bar

3. Fig. 4 - seems to be fuzyy. Moreover, x,y axes need to be described

4. Fig.  8 - please fit window to a data range

5. Fig. 16-17 - please add units on y-axis

6. A valuable will be any statistical data analysis

7. Did the Authors try to validate this model for other media/Reynolds numbers? Please comment limitations of the presented approach. 

8. In the paper exist a lot of abbreviations and symbols - please add a section - nomenclature

Author Response

Reviewer 2

The paper is generally well written and contains valuable results. As a Reviewer, I would like to express my positive opinion Before publication, please follow the points listed below.

  1. Please explain all symbols in existed Equations, like Eq1

Change1: The symbols in Eq1 are explained

Original Version

(1)

Current Version

(1)

The Re is Reyolds number. The ρ and v are the density and velocity of fluid. The d and μ are the hydraulic diameter of the fracture and the viscosity.

  1. 3 - PLease add scale bar

Change1: The scale bar is adeed.

Original Version

Fig.3 Outcrop sample and duplicated fractures

Current Version

Figure 3. Outcrop sample and duplicated fractures.

  1. Fig. 4 - seems to be fuzyy. Moreover, x,y axes need to be described

Change1: The Figure has been changed and the x, y axes are described.

Original Version

Figure 4. Generated mesh and mesh quality.

The total generated mesh number of the fracture is 151392 shown in Fig.4 and all of the mesh meet the calculation requirements.

Current Version

The total generated mesh number of the fracture is 151392 shown in Fig.4 and all of the mesh meet the calculation requirements. The mesh quality is represented on the x-axis, when the x values closer to 1 indicating better mesh quality. The y-axis represents the number of grid cells, and the upward blue arrow in the figure indicates that the actual number of meshes is much larger than the upper limit of the coordinate.

  1. Fig.  8 - please fit window to a data range

Change1: The data range has been revised.

Original Version

(a) v=12m/min

(b) v=24m/min

(c) v=36m/min

Current Version

(a) v=12m/min

(b) v=24m/min

(c) v=36m/min

  1. Fig. 16-17 - please add units on y-axis

Change1: The y-axis represents the percentage of sand bank height in the fracture.

Original Version

Figure 16. Changes of sand bank equilibrium height with slurry velocity and sand diameter

Current Version

Figure 16. Changes of sand bank equilibrium height with slurry velocity and sand diameter (The y-axis represents the percentage of sand bank height in the fracture.)

  1. A valuable will be any statistical data analysis
  • This paper establishes a regression model for the prediction of proppant migration and we studied the migration law of proppant with the flow rate, sand ratio and particle diameters. and clearly shows the migration and proppant law of proppant in rough fractures. And the statistical data analysis was not considered.
  1. Did the Authors try to validate this model for other media/Reynolds numbers? Please comment limitations of the presented approach. 
  • We have tried the proppant migration in the smooth fracture model, and the numerical results are in good agreement with the experimental results. The current limitation is that it is difficult to achieve consistency with the real fracture surface morphology for research.
  1. In the paper exist a lot of abbreviations and symbols - please add a section - nomenclature

Change1: The nomenclature has been added after the References.

Original Version

None

Current Version

Nomenclature

: the stress tensors of the fluid

: the stress tensors of the solid

: the acceleration of gravity

: velocity of fluid

: velocity of solid

: granular temperature

: volume fraction of fluid

: volume fraction of solid

: density of the fluid

: density of the solid

Cd: the drag coefficient

d: hydraulic diameter

ds: the diameter of the particle

g0: radial distribution function

P: pressure of the all phases

Re: Reyolds number

Res: Reynolds number of soild

v: velocity

β: the Gidaspow drag force coefficient

μ: viscosity

μs,col: collisional viscosity

μs,fri: fractional viscosity

μs,kin: kinetic viscosity

μs: shear viscosity of solid

ρ: density

Thank you again for your suggestions and is very useful for improving our manuscript.

Reviewer 3 Report

The development of effective technologies for providing flow channels for oil and gas production is very important for the exploitation of unconventional resources. The authors of the paper “Impacts of fracture roughness and near-wellbore tortuous on proppant transport within hydraulic fractures” have shown an increase in sand bank equilibrium height with increases in the fracture roughness using finite element simulation. The authors have presented a significant number of calculated results. However, some points of the paper are needed to be explained in more detail accordingly following comments:

1. Suri et al in Ref. 20 have already shown the influence of the surface roughness on the proppant transport in hydraulic fractures. The authors should highlight the scientific novelty of their findings in comparison with previous investigations.

2.  The authors have defined the fracture surface roughness as the set of the balls with the radius corresponding to Ra. However, this model seems to be too simplified. The real surface does not correspond to the model in the ANSYS. It is unclear why the authors did not use the real surface in their simulation if they make 3D-scanning of the object and have it 3D-model.

3. It is not evident from Figure 20 that with the increase of the roughness, the height of the sand bank gradually increases. From the figure, the only decrease in the moving forward rate of the sand bank in the fractures is seen. The authors should provide additional approvement of the statement.

4.  Minor corrections:

- The scale in Figure 8 should be decreased to 10 cm.

- The Ra in Figure 6 should be as a radius not as a diameter.

Author Response

Reviewer 3

The development of effective technologies for providing flow channels for oil and gas production is very important for the exploitation of unconventional resources. The authors of the paper “Impacts of fracture roughness and near-wellbore tortuous on proppant transport within hydraulic fractures” have shown an increase in sand bank equilibrium height with increases in the fracture roughness using finite element simulation. The authors have presented a significant number of calculated results. However, some points of the paper are needed to be explained in more detail accordingly following comments:

  1. Suri et al in Ref. 20 have already shown the influence of the surface roughness on the proppant transport in hydraulic fractures. The authors should highlight the scientific novelty of their findings in comparison with previous investigations.
  • The novelty of our study is mainly reflected in the study of the forward rate of sand bank in rough fractures. The sand bank formation in tortuous fractures is further studied.
  1. The authors have defined the fracture surface roughness as the set of the balls with the radius corresponding to Ra. However, this model seems to be too simplified. The real surface does not correspond to the model in the ANSYS. It is unclear why the authors did not use the real surface in their simulation if they make 3D-scanning of the object and have it 3D-model.
  • 3D scans are used to obtain the fracture surfaces features. Then, Ks is used to characterized the roughness of fracture. Then, changing Ks in ANSYS-Fluent is consistent with the Ra of the fracture. And it is found that the formed sand bank is close to the experimental result. After that, we directly changed the Ks embedded in the wall boundary condition to study the effect of wall roughness on sand bank formation and sand transport.
  1. It is not evident from Figure 20 that with the increase of the roughness, the height of the sand bank gradually increases. From the figure, the only decrease in the moving forward rate of the sand bank in the fractures is seen. The authors should provide additional approvement of the statement.
  • Figure 20 mainly shows the relationship between the sand bank forward rate and roughness. Figure 18 shows the effect of roughness on the height of the sand bank and the shape of the sand bank
  1. Minor corrections:

- The scale in Figure 8 should be decreased to 10 cm.

- The Ra in Figure 6 should be as a radius not as a diameter.

Change1: The data range has been revised.

Original Version

(a) v=12m/min

(b) v=24m/min

(c) v=36m/min

Current Version

(a) v=12m/min

(b) v=24m/min

(c) v=36m/min

Change2: The Ra in Figure 6 has been revised.

Original Version

Current Version

Thank you again for your suggestions and is very useful for improving our manuscript.

Round 2

Reviewer 1 Report

Thanks for answering my questions and accommodating some of my suggestions. Minor language corrections are needed in some of the new sections that have been added. For example, Page 2, lines 80/ 81.

Also, you need to clean up the references section with redundant reference citation (page 22, between #25 and #26).

Author Response

Reviewer 1

Thanks for answering my questions and accommodating some of my suggestions. Minor language corrections are needed in some of the new sections that have been added. For example, Page 2, lines 80/ 81.

Also, you need to clean up the references section with redundant reference citation (page 22, between #25 and #26).

Change1: The language has been modified.

Original Version

Compared with smooth fracture, rough wall allows more proppant deposited in a cell at high pumping rate

Current Version

Compared with smooth fracture, there will be more proppant deposited in rough fractures at a high pumping rate.

Change2: The references have been cleaned up.

Original Version

Gale, J. F. W., Elliott, S. J. & Laubach, S. E. Hydraulic Fractures in Core From Stimulated Reservoirs: Core Fracture Description of HFTS Slant Core, Midland Basin, West Texas. in Proceedings of the 6th Unconventional Resources Technology Conference (American Association of Petroleum Geologists, 2018). doi:10.15530/urtec-2018-2902624.

Current Version

Gale, J. F. W., Elliott, S. J. & Laubach, S. E. Hydraulic Fractures in Core From Stimulated Reservoirs: Core Fracture Description of HFTS Slant Core, Midland Basin, West Texas. in Proceedings of the 6th Unconventional Resources Technology Conference.

Thank you again for your suggestions and is very useful for improving our manuscript.

Reviewer 3 Report

The authors have answered previous comments. However, the scale Y in Figure 8 is still not appropriate. The authors have changed the font's size but should change the scale from 10 to 30 cm.   

Author Response

The authors have answered previous comments. However, the scale Y in Figure 8 is still not appropriate. The authors have changed the font's size but should change the scale from 10 to 30 cm.

Author reply:

In this study, the fracture height is 30 cm, by using the scale which can clearly show the sand bank shape in the fractures.

Thank you again for your suggestions and is very useful for improving our manuscript.